# Synthesis of Human Phase I and Phase II Metabolites of Hop (*Humulus lupulus*) Prenylated Flavonoids

**DOI:** 10.3390/metabo12040345

**Published:** 2022-04-12

**Authors:** Lance Buckett, Sabrina Schönberger, Veronika Spindler, Nadine Sus, Christian Schoergenhofer, Jan Frank, Oliver Frank, Michael Rychlik

**Affiliations:** 1Analytical Food Chemistry, Technical University of Munich, Maximus-von-Imhof Forum 2, 85354 Freising, Germany; sabrina.schoenberger@tum.de (S.S.); veronika.spindler@tum.de (V.S.); michael.rychlik@tum.de (M.R.); 2Department of Food Biofunctionality (140b), Institute of Nutritional Sciences, University of Hohenheim, Garbenstraße 28, 70599 Stuttgart, Germany; nadine.sus@nutres.de (N.S.); jan.frank@nutres.de (J.F.); 3Department of Clinical Pharmacology, Medical University of Vienna, Währinger Gürtel 18-20, 1090 Vienna, Austria; christian.schoergenhofer@meduniwien.ac.at; 4Food Chemistry and Molecular Sensory Science, Technical University of Munich, Lise-Meitner-Str. 34, 85354 Freising, Germany; oliver.frank@tum.de

**Keywords:** metabolites, synthesis, prenylated flavonoids, hops, beer, xanthohumol, blood analysis

## Abstract

Hop prenylated flavonoids have been investigated for their in vivo activities due to their broad spectrum of positive health effects. Previous studies on the metabolism of xanthohumol using untargeted methods have found that it is first degraded into 8-prenylnaringenin and 6-prenylnaringenin, by spontaneous cyclisation into isoxanthohumol, and subsequently demethylated by gut bacteria. Further combinations of metabolism by hydroxylation, sulfation, and glucuronidation result in an unknown number of isomers. Most investigations involving the analysis of prenylated flavonoids used surrogate or untargeted approaches in metabolite identification, which is prone to errors in absolute identification. Here, we present a synthetic approach to obtaining reference standards for the identification of human xanthohumol metabolites. The synthesised metabolites were subsequently analysed by qTOF LC-MS/MS, and some were matched to a human blood sample obtained after the consumption of 43 mg of micellarised xanthohumol. Additionally, isomers of the reference standards were identified due to their having the same mass fragmentation pattern and different retention times. Overall, the methods unequivocally identified the metabolites of xanthohumol that are present in the blood circulatory system. Lastly, in vitro bioactive testing should be applied using metabolites and not original compounds, as free compounds are scarcely found in human blood.

## 1. Introduction

Prenylated flavonoids (PF) are secondary plant metabolites found in a range of plants, notably amongst the families Cannabaceae (hops, *Humulus lupulus*), Moraceae (mulberry, *Morus Albas*), and Fabaceae (liquorice, *Glycyrrhiza. glabra*), some of which are part of the human diet [1,2]. Many prenylated flavonoids are also found in the non-food components of plants, such as mulberry trees; the highest concentrations of prenylated flavonoids are in roots and leaves [3]. Prenylated flavonoids play a role in plant defence, as they perform a range of anti-mycotic, anti-bacterial, and anti-viral activities, which is perhaps why plants invest heavily in their biosynthesis [4]. Arguably the most frequently researched prenylated-flavonoid derivative is xanthohumol (XN), a chalcone from hops, which is a main ingredient in beer production. As beer is the most heavily consumed alcoholic beverage, XN and related compounds are believed to originate solely from beer in a dietary context [5,6,7]. The lipophilic prenyl group of XN seems to increase the bioavailability of prenylated chalcone compared to non-prenylated chalcone, which may also improve their efficacy compared to unprenylated flavonoids [8].

However, comparatively little is known about how prenylated flavonoids influence health, even though emerging evidence suggests that they have potential for the prevention and/or therapy of a range of disorders, including those associated with the metabolic syndrome [9]. The health implications of XN consumption have been mainly studied in rodents and are largely unknown in humans [9]. In rats, for example, alcohol-induced liver damage was reduced by supplementation with XN, most likely due to its antioxidant activity [10]. XN slows cell division in certain cancer cells, e.g., the MCF-7, HT-29 and A-2780 cell lines [11]. Xanthohumol-C (XN-C), an analogue of XN, has shown potential in neuronal regeneration through the upregulation of human doublecortin, which was demonstrated in a transfected mouse embryo fibroblast model [12]. Along with XN, isoxanthohumol (IXN) has been suggested as a potential treatment for COVID-19 (SARS CoV-2) [13,14,15]. XN is also being used in clinical trials as a preventive measure against Crohn’s disease, and it might help relieve symptoms [16]. Furthermore, XN is pro-oestrogenic as it can spontaneously cyclise into the isomer IXN, which is further demethylated by gut bacteria, resulting in the formation of 8-prenylnaringenin (8-PN), a renowned phytoestrogen [7]. Hop extracts were historically used as a treatment for gynaecological disorders and to relieve women of menopausal symptoms. Currently, for improving the latter, there are some hop products on the market [17].

Previous research focused on free compounds’ bioactivities, while metabolites have been largely ignored. Once XN is absorbed into metabolically competent cells of the intestine, it is rapidly metabolised, transformed into conjugates, and partly re-secreted into the gut. Some of the XN and its metabolites are delivered to the portal blood and, thus, reach the liver, where they are further metabolized and either excreted via bile or secreted into the circulation, from where they may reach either their “site of action” or the kidneys, for renal elimination (Figure 1). Thus, the free (unmetabolised) form does not actually reach the cells. 

We therefore investigated the synthesis of XN metabolites and compared them to the metabolome of a human blood sample in order to provide the relevant human metabolites of XN for future experiments aimed at investigating their biological activities. 

## 2. Results and Discussion

The synthetic approaches to obtaining the analytical standards of certain metabolites of XN via the Königs–Knorr method are presented in Figure 2, including the most abundant metabolites, XN-7-O-glc and IXN-7-O-glc [18]. Additionally, we carried out in vitro glucuronidation by incubating the XN with pig-liver microsomes to further confirm the synthesised products. To achieve the hydroxylation of the prenyl group of PF while retaining the prenyl character, a Riley oxidation process was selected [19]. The sulfation used a reaction based on the work of Legette et al. (2014) that involves base acid sulfation, although the authors did not isolate the products [20]. Currently, no reference standards obtained via synthetic methods are generally available and, therefore, new methods of isolation of vast quantities of PF for synthesis make these approaches viable, as XN isolated from spent hops is more obtainable [16,21]. Understanding the metabolism will give further evidence of how these compounds interact at a biological and cellular level in vivo. A range of XN products is already available on the market and certain pharmaceutical formulations are being developed and researched, even though the exact mechanism through which they influence health has not been elucidated. Lastly, XN is already being developed in the form of bioavailable gold nano composites used in crop protection and as a food preservative [22,23,24,25].

The synthetic metabolites were characterised by HPLC, MS, LC-MS/MS, and NMR. The glucuronides were subjected to an additional enzymatic deconjugation using β-glucuronidase from *Helix pomatia* to supplement the NMR data and confirm that they were in the β-anomer configuration. The method used to chemically and biochemically (without the need for ultracentrifugation) produce some metabolites of hop prenylated flavonoid XN was successful, which might be useful for bioactivity testing and analytical methods in future work. The MRM mass spectrometry showed that the products fragmented in the same way as was observed in other in vivo studies investigating XN or related PF metabolism (see Appendix A) [20,26,27,28]. 

### 2.1. Hydroxylation of the Prenyl Group

The hydroxylation of the prenylated flavonoids showed that the Riley oxidation was successful and resulted in yields of between 10–25% in compounds **5**–**9**. Upon the initial analysis by mass spectrometry, the hydroxylated products all revealed an additional mass increment of 14 Da, which supported the assumption that the respective aldehydes formed (no further analysis was carried out). To reduce the aldehyde into hydroxyl, NaBH_4_ was used, which resulted in a mass increment correlating to hydroxylation (+16). The UV spectra of compounds **5** and **6** each showed an absorption maximum of λ = 370 nm and 7–9 of λ = 290 nm, indicating that the products were in chalcone and flavanone form, respectively. Compounds **5**–**7** revealed an *m/z* signal at 369, while 8 and 9 showed an *m/z* signal at ~355, which suggests that the mass of a hydroxyl group (+16) was added to all the compounds. Thus, the alkene moiety of the prenyl group was retained. The MS analysis of the hydroxylation of the XN revealed two major products, as two peaks were separable in the chromatogram (see Figure 3). Once the NMR (NOESY) data were collected, they revealed that the two products were the *E* and the *Z* isomer, respectively. It is believed that trace amounts of the *Z* isomer were produced in 7–9, but none of the separation methods were able to resolve. Substances 8 and 9 produced a small shoulder, which can be visualised in the LC-MS/MS chromatogram. 

The NMR analysis further confirmed the hydroxylation of a terminal methyl group (4′’ or 5′’) of the prenyl moiety as the respective former signal was shifted to higher frequencies (between 3 ppm and 4 ppm, depending on the PF hydroxylated), at which point the integral was two instead of three protons in all of the analysed PFs. To confirm the stereochemistry of the products, 2D NMR experiments were carried out. The absolute confirmation was determined using a NOESY experiment via a correlation between the H2′’ and the two remaining protons on the hydroxylated methyl in *E* form and the H2′’ and the methyl group in *Z* form [29]. Small secondary products that were visible in the same MRM mass chromatograms were also seen in the NMR spectra for the flavanone form, and it was assumed that they were produced, although they could not be isolated in large enough quantities in the work presented here.

Hydroxylated PFs 5–7 have only been found by in vitro incubation either with human liver microsomes or through metabolism using yeast *P. membranifaciens* [26,30]. This is the first report on the chemical synthesis of these compounds. 

### 2.2. Synthesis of Sulfates

The sulfation of the prenylated flavonoids was successful, but was not regiospecific (yields between 50–60%). Protection chemistry, e.g., hydroxyl protection using MOM-Cl, would potentially enable a more specific conjugation, but regardless of whether one isomer is created, the other can be elucidated via the assumption that MS fragmentation patterns are similar in their identification of isomers. It was assumed that no desmethylxanthohumol derivatives formed as they were unlikely to be present due to their high instability in acidic environments, such as the stomach. However, for the quantification of the other metabolites, isolated isomers of each compound would be necessary. 

### 2.3. Biosynthetic Glucuronidation

In our initial attempts to produce glucuronides, we used biochemical methods. This was sufficient for the production of analytical standards, but it was costly when considering the amounts (in mg) required for cell culture incubation experiments. This was mainly due to the high cost of the cofactor uridine diphosphate glucuronic acid and the antibiotic alamethicin; therefore, this approach was abandoned after the successful production of the XN-7-O-Glc and XN-4′-O-Glc. The advantage of biosynthesis is that if tentative evidence is required, it can be performed quickly, and using this approach enabled the confirmation of the XN-C-glc and IXN-C-glc. Moreover, it provided evidence about the selectivity of which isomer was produced as the synthetic glucuronidation only resulted in one major product. Only mixed Uridine 5′-diphospho-glucuronosyltransferases (UGT) were used and, perhaps, further engineering might lead to increased yields. Inadvertently, our results suggested that the enzymes from the pigs glucuronidated XN in a similar way to humans, favouring XN-7-O-glc production versus XN-4′O-glc [31].

#### XN-C-Glucuronide and IXN-C-Glucuronide

All the synthetic methods on the pyranone PF failed to produce enough material for an NMR, although trace amounts of xanthohumol-C glucuronide were found after the Königs–Knorr approach. Therefore, it was assumed that this is a major metabolite and only tentative evidence was obtained for the formation of XN-C and IXN-C-glucuronide. Therefore, glucuronic acid was assumed to be attached to the phenol group. This regiospecificity was evidenced by the crystal structures of the XN and IXN, revealing a strong hydrogen bond between the keto and 5-hydroxy moieties, which directs the conjugating enzyme UGT to the phenol group. Instead, any enzymatic conjugation at the 5-hydroxyl would need to overcome the energy used for displacing the hydrogen bond before the occurrence of metabolism [32].

### 2.4. Synthesis of Glucuronides

Initially, the Königs–Knorr reactions on the XN, as the most abundant PF in the hops, were investigated, and the conjugation was confirmed by MS (data not shown). However, deprotection, using a variety of established methods, resulted in either the destruction of the starting material or no removal of the protective acetate groups on the glucuronic acid methyl ester. Additionally, the use of KCN only resulted in the removal of the acetate groups, but failed to remove the methyl ester, making it a non-viable method for the deprotection of PF-glucuronides within the Königs–Knorr approach. Therefore, to obtain the XN-7-O-glc and IXN-7-O-glc, acetate methyl ester was first synthesised and then deprotected by the addition of toluene sulfonic acid, resulting in the conversion of the IXN-7-O-glc into XN-7-O-glc. It seemed that the conversion between flavanone and chalcone could be partially controlled by the addition of KOH, most likely resulting in ring opening, and neutralisation allowed the formation of XN-7-O-glc along with IXN-7-O-glc, depending on the amount of KOH used. However, the reaction required the careful addition of KOH which, if added in excessive amounts, could have resulted in the destruction of the glucuronide conjugate. The same approach was applied to the 8-PN and 6-PN, but based on the observation that both 8-PN-7-O-glc and 6-PN-7-O-glc were produced, it was not deemed necessary to isolate the pure 6-PN and 8-PN. Instead, a mixture was sufficient for our analytical purposes, thus avoiding a time-consuming purification step (yield 12%). The application of Königs–Knorr on XN as a starting material resulted in only trace amounts of the glucuronide product. Therefore, IXN, which produced a yield of 21.3%, is a better starting material.

### 2.5. LC-MS/MS of Synthesised Analytes

In non-targeted tandem mass spectrometry, data-dependent acquisition (DDA) is a common method used to fragment a (user-defined) number of the highest ion count “masses” from the MS1 scan within certain time frames. The MS2 scans correspond to the MS1 scan, which simplifies data interpretation and was applied to analyse the synthetic products in Figure 3. The DDA of the standard mixture revealed the fragmentation of each compound corresponding to the literature and the expected fragmentation proposed therein (see Appendix A for details) [28].

### 2.6. LC-MS/MS of Analytes in Human Blood after Xanthohumol Consumption

The selection of the blood sample 1 h after dosing was based on the work of Legette et al. (2014), who demonstrated peak XN concentrations after 1 h of consumption [20]. Therefore, blood plasma samples were taken randomly from a human trial at the 1-h time point. Upon the initial analysis, the mass spectrum was saturated with components from the human plasma (as components from the blood were selected over the XN metabolites), making DDA unsuitable for human plasma; therefore, data-independent acquisition (DIA) was carried out. Contrary to the DDA, the DIA fragments of all the ions in the MS1 spectra retaining all information on all the detected compound fragments in the spectra, although it was difficult to determine the precursor for each molecule. In addition, the standard mixture was also determined using the DIA for co-chromatography and the mass-spectrometry identification of the fragmentation of the compounds in the human blood plasma (*n* = 2). Regarding the glucuronides, all the synthesised metabolites were found in the human plasma sample; therefore, the positions of the glucuronic acids were identified unequivocally. Additionally, the compounds IXN-4′-O-Glc, 8-PN-4′O-Glc, and 6-PN-4′O-Glc were tentatively identified as they possessed the same nominal mass and fragmentation pattern as the synthesised compounds (Figure 4). This coincides with the assumption of Yilmazer et al. (2001) regarding the analysis of glucuronides after enzyme incubation and confirms that they were correct, due to their identification of the same elution order on a C18 column [33].

Unlike the glucuronides, for the XN sulfates in the human plasma (Figure 5), only XN-4′-O-sulfate was identified unequivocally as no other co-chromatographing peak was found. As being tentatively identified at level 3 (Table 1) according to Schymanski et al. (2014), two other products were found, namely IXN-7-O-sulfate and XN-7-O-sulfate [34]. No IXN-4′-sulfate was found, and the most prevalent product by intensity was, tentatively, XN-7-O-sulfate.

In Figure 6, the sulphated prenylnaringenins revealed that, out of the synthesised compounds, only compound **14** (6-PN-7-O-sulfate) was found, along with the tentative compounds 8-PN-7-O-sulfate (compound **12**) and 6-PN-4′O-sulfate (compound **15**).

The description of XN metabolism is much more complicated than XN reaching cells and inducing certain bioactivities. Once consumed, XN is modified by a series of processors, including passive and non-passive routes, resulting in the formation of many metabolites (Figure 1). Furthermore, the gut microbiota plays an important role [35]. Another factor that makes investigating the bioactivities of XN challenging is that most in vivo studies of bioactivity use mouse models. The in vivo mouse model for XN is prone to error as most mice do not have the human microbiota that are essential to study the bioactivity effects of XN supplementation [36,37,38]. However, some of the metabolites are probably bioactive as in vivo effects are reported from XN consumption [39,40]. Some metabolites might be rapidly excreted, which would suggest that their absorption into cells is limited, although to answer this unequivocally, a preliminary investigation using synthesised metabolites with a Caco-2 absorption assay is necessary [9,41]. As very little free XN and PF are found in human samples after XN consumption, testing the bioactivities of synthesised metabolites in vitro would provide further evidence of what might occur in vivo. A summary of the compounds found in the blood sample is shown in Table 1. No hydroxyls were found in the blood plasma sample (see Appendix A for the chromatograms). Hence, large-scale intervention trials should be carried out to determine whether hydroxylated prenylflavonoids are produced in rare cases. Therefore, the production of these metabolites is essential for future work on XN metabolism. Additionally, it might be that the dose of XN was not sufficient to produce sufficient quantities of t hydroxylated metabolites, or that they are quickly conjugated by phase II enzymes and thus not sufficiently long-lived. Another, less common use of the synthesised metabolites is the investigation of the concentration of metabolites in the environment, as free XN is known as a pollutant and its metabolite haves not been investigated [42].

## 3. Materials and Methods

### 3.1. General 

All reactions were carried out using analytical-grade solvents unless otherwise stated. Biochemical reactions were conducted at 39 °C in an orbital shaking bath (GFL 1092, Burgwedel, Germany).

### 3.2. Reference Compounds 

Xanthohumol, xanthohumol-C, isoxanthohumol, isoxanthohumol-C, 6-prenylnaringenin, and 8-prenylnaringenin from hops were synthesised previously [43].

#### 3.2.1. General Reaction of Hydroxylated Prenylated Flavonoids

To hydroxylate the prenylated flavonoids, a Riley oxidation method was employed [19,44]. A reference prenylated flavonoid (1 molar e.q) was dissolved in 3 mL ethanol. Along with SeO_3_ (1.2 molar equivalence), the reaction was taken to reflux for 48 h and quenched by the addition of ethyl acetate and water; subsequently, liquid–liquid extraction was carried out by washing the organic phase with water (25 mL) three times. This was employed to 6-PN, 8-PN, IXN, and XN individually. The organic phases of each reaction without further clean-up were further reacted with NaBH_4_ at 1.2 molar equivalence (of the aldeyde) by dissolving the reactant and product of the oxidation in ACN and leaving it to stir for 30 min. The products were then subjected to liquid– liquid extraction again and washed over celite 454, concentrated and purified using preparative HPLC, and submitted to NMR, HPLC, LC-MS/MS and HR-MS analysis. 

#### 3.2.2. Synthesis of Sulphate Conjugates 

In an adaption from Legette et al. (2014) [20], the general scheme was as follows. Individually selected prenylated flavonoids (50 mg) were dissolved in pyridine 1 (mL) and, subsequently, 80 µL of Chlorosulfonic acid was added, while the flask was on an ice bath. The reaction was left to stir vigorously for 30 min, followed by stirring overnight at room temperature. The following day, the reaction was quenched with the slow addition of approximately 20 mL of NaCO_2_ until the solution stopped fizzing. All samples were submitted to liquid–liquid extraction by washing the water with 5 × ethyl acetate (EtOAc), the EtOAc fractions were washed with brine, and the rotation evaporator was set at 60 °C to remove most of the pyridine. The samples were columned through Sephadex LH-20 using MeOH for elution, concentrated, and then purified by preparative HPLC [43]. 

#### 3.2.3. Synthesis of Synthetic Glucuronic Acid Conjugates

Synthesis of the glucuronic acid donor was carried out according to Jongkees and Withers (2011) in a dark environment [18]. To IXN, in a dry flask in argon atmosphere, 20 mg of IXN was dissolved in dry acetonitrile (dried with molecular sieves, Sigma Aldrich, Saint Louis, MO, USA) followed by the addition of 27 mg (1,2 eq) of acetobromo-α-d-glucuronic acid methyl ester. The reaction was left to stir for 5 min before AgO_2_ (2,5 eq, 32.4 mg) was added. The reaction was left at room temperature for 20 h before being filtered through a celite plug, followed by silica gel (mesh 80–200), filtered through filter paper, and concentrated. The reaction product was resuspended in EtOAc and washed with H_2_CO_3_ twice, with water twice, and with brine once before being concentrated. Deprotection was conducted by the addition of MeOH/H_2_O, changing the pH to 12, and leaving the product to stir for a further 3 h, before performing liquid–liquid extraction. Lastly, the obtained suspension was filtered through celite, and the filtrate was columned with Sephadex LH-20 and prepared with preparative HPLC according to previous work [43]. To obtain 6-PN-7-O-Glc and 8-PN-7-O-Glc, a mixture of 6-PN and 8-PN was used as starting material. 

### 3.3. Biosynthesis of Glucuronic Acid Conjugates

#### 3.3.1. Purification of Pig-Liver Microsomes

Enzymatic synthesis of glucuronide conjugates was performed by using pig liver. Liver microsomes were prepared according to a modified version of the method of Kamath et al. (1971) [45]. The liver (from a pig slaughtered less than 1 h prior) was a gift from a local butcher. The liver was transported on ice and immediately homogenised at a ratio of 4 mL to 1 g tissue with a 10% sucrose buffer containing 0.05 M tris HCl, 150 mM KCl, 0.005 M MgCl_2_, 2 mM EDTA, and 1 mM dithiothreitol at pH 7.5. The liver homogenate was then centrifuged at 10,000× *g* for 15 min at 4 °C. Afterwards, the supernatant was removed and collected in a centrifuge tube. The supernatant was diluted with 25 mL of cold 0.0125 M sucrose solution containing 0.008 M CaCl_2_ and 0.005 M MgCl_2_, and was stirred a few times on ice. The solution was again centrifuged but at 1500× *g*, and at 4 °C, for 10 min. The pellet that formed was collected and rinsed with a 15-mL, 10% sucrose solution. The process was repeated twice, but each time the pellet was dispersed (vortexed) in fresh sucrose (10%). Finally, the pellet was solved in a 0.05 M Tris-HCl buffer containing 150 mM KCl, 0.005 M MgCl_2_, 2 mM EDTA, and 1 mM dithiothreitol pH 7.5. Aliquots (1.5 mL) were flash-frozen in liquid N_2_ and stored at −80 °C until usage. 

#### 3.3.2. Glucuronidation of Flavonoids Using Pig-Liver Microsomes

A method similar to that described by Ladd et al. (2016) [46] was applied. A substrate of 2 mM was dissolved in ethanol (VWR) along with 1 mM alamethicin (Cayman Chemical Compound); the solution mixture was incubated in a vial at 39 °C for 17 h and quenched by the addition of 1 mL of cold MeOH, and the sample was centrifuged at 10,000× *g* 4 °C for 5 min. The sample was washed again and recentrifuged. The supernatants were combined and dried under N_2_, reconstituted in 90:10 ACN:MeOH, and analysed or prepared by HPLC. After collecting the peaks from HPLC, the purified compounds underwent LC-MS/MS analysis and, of these compounds, XN-Glc1 and XN-Glc2 underwent NMR analysis. 

#### 3.3.3. Protein Content of Liver Microsomes

The protein content of the liver microsomes was estimated using Bradford assay and bovine serum albumin (BSA) for calibration. The reference protein BSA (lyophilized powder, crystallized, ≥98.0% GE) was made up to 1 mg/mL and diluted in a potassium phosphate buffer (pH 7) to corresponding concentrations. Calibration graphs were produced according to [47]. 

#### 3.3.4. Enzymatic Digestion of Glucuronides

To confirm the β-anomer of the synthesised glucuronides, they were treated with β-glucuronidase from *Helix pomatia* (G7017, Sigma Aldrich, Saint Louis, MO, USA). A method derived from Grebenstein et al. (2017) [48] was slightly modified. The synthesised glucuronides (dissolved in ethanol) were treated with 3.4 µL of the β-glucuronidase (≥100.000 units/mL), suspended in a buffer (NaOAc 2 mM at pH 5.1) and incubated at 37 °C for 2.5 h [48]. A control was used by adding the glucuronide conjugate, but without adding any enzymes. The samples then underwent HPLC, LC-MS/MS, and NMR analysis. 

### 3.4. Analysis of Samples

#### 3.4.1. HPLC

Analytical HPLC was carried out using a Shimadzu LC-DAD system (Shimadzu, Kyoto, Japan; Degaser: DGU-20A3R prominence degassing Unit, Liquid Chromatograph: LC-20 AD prominence liquid chromatograph, SIL-20A HAT prominence autosampler, and SPA-M20A prominence diode-array detector) with a YMC-Pack Pro C18 column (S-5 µm, 12 nm, 150 × 10.0 mm). The preparative HPLC was performed according to Buckett et al. (2020) [43]. Regarding the biosynthesised compounds, each peak was collected using the analytical method and, after approximately 20 runs, this resulted in enough material for NMR. 

#### 3.4.2. Mass Spectrometry

MRM fragmentation of all analytes was determined by injecting 1 × 10^−4^ mg/L of the synthesised reference compounds directly using an autosampler (Shimadzu, nexera 2 system). The time-of-flight mass spectrometer (Shimadzu 9030) was tuned beforehand using NAI (Shimadzu) with a resolution above 40,000 at 1625.8 *m/z* in 2 kHz mode, and data-dependent acquisition (DDA) mode was selected from a range of 100–1000 Da with a threshold set at 1000 cps. The nebulizing gas flow was set at 3.0 L/min, the heating gas flow at 10 L/min, the interface temperature at 300 °C, and the desolvation temperature at 526 °C. The interface had a voltage of −40 kv, drying gas of 10 L/min, DL temperature of 250 °C, and heat block of 400 °C. 

#### 3.4.3. Nuclear Magnetic Resonance Spectroscopy

The 1D-/2D-NMR experiments for structure elucidation were performed on a 500 MHz or 600 MHz Bruker AVANCE NEO system, equipped with a triple-resonance cryo-probe (TCI, Bruker, Rheinstetten, Germany) at 300 K. Samples were either dissolved in 600 µL Methanol-d_4_ or Acetone-d_6_, and chemical shifts are reported in parts per million (ppm) relative to solvent signals in the ^1^H NMR and 13C NMR spectra. All pulse sequences were taken from the standard Bruker pulse sequence library. Data processing was performed with Topspin software (version 4.1.1) or MestReNova 14.0 (Mestrelab Research, Santiago de Compostela, Spain).

#### 3.4.4. LC-MS/MS

The synthesised compounds at approx. 2.5 µg/L and blood samples were injected (5 µL) using the following parameters and subjected to analysis with a Shimadzu Nexera UHPLC system coupled with a Shimadzu 9030 Q-ToF mass spectrometer (Kyoto, Japan). A Waters BEH c18 100 × 2.1 mm, 1.7 µm separated the analytes with acidified solvents using 0.1% formic acid, H_2_O (solvent A), and ACN (solvent B). The gradient started at 20% B and ran isocratically for 4 min at a flow rate of 0.4 mL/min; it then slowly increased using a gradient at 85% B at 10.5 min and was held for 1.5 min until it was ramped up to 99% B for 2.5 min before returning to 20% B over 1 min and an equilibration time of 3 min, giving a total measurement time of 20 min. The column oven temperature was set at 40 °C. Due to the saturated mass spectra, blood sample underwent a data-independent acquisition mode to collect data. The data-independent analysis (DIA) window was set at 100–800 *m*/*z* and each MS/MS event over 20 Daltons, while CE was set at 35 V with spread of 20 V. One event was reserved for MS1 spectra and at the end of each run, a reference material (Agilent low-mass mix) was introduced using the CDS. Each run was calibrated to these masses. A blood sample from a random study participant was taken after 60 min of consumption of micellar XN (43 mg), which was determined to contain the highest amount of glucuronide conjugates (data not shown). The blood sample was directly subjected to analysis. The study included healthy volunteers and was conducted at the Department of Clinical Pharmacology, Medical University of Vienna, between October 2018 and August 2019. The independent ethics committee approved the study before its initiation (ethics number 1580/2018) and all participants gave their oral and written informed consent. In short, healthy volunteers received three capsules of native or micellar XN (43 mg) in a randomized, double-blind crossover manner for seven days. The main objectives were to determine the pharmacokinetics and compare the pharmacodynamics of two XN formulations with specific focus on anti-inflammatory effects (data not published).

### 3.5. Preparation of the Blood Sample

The blood samples were prepared according to van Breemen et al. (2020), although an additional 10 µL of 1 M ascorbic acid was added before centrifugation and the samples were resolved in 50 µL of 90:10 ACN:MeOH [27,48].

## 4. Conclusions

With the methods presented here, many phase I and phase II metabolites of XN were generated. The synthesised compounds represent common metabolites, such as phase I hydroxyls and phase II conjugates, namely, glucuronides and sulfates. This includes the most predominant glucuronide conjugates of XN. The identity of the compounds allowed the confirmation of the functional groups’ positions. Due to the production of chemical standards in high enough amounts to perform an NMR analysis, an unequivocal characterisation was accomplished and a tentative identification of the analytes found in human blood, without the use of isolated standards, was obtained. The Riley oxidation was not stereospecific towards XN and produced two isomers. In the analysis of the blood, no hydroxylated compounds, nor their conjugates, were found; this does not rule out their presence, but it does reveal that they might be concentrated at extremely low levels or not produced at the dose of XN that was given (43 mg). Furthermore, producing one or two different conjugates allows through the process of elimination to identify functional group positions without producing the analytical standard. The bioavailability of these compounds needs to be addressed in future investigations to understand the mechanism behind XN supplementation, as it is currently undergoing clinical trials and is a leading candidate natural product in the treatment of COVID-19 infection.

## Figures and Tables

**Figure 1 metabolites-12-00345-f001:**
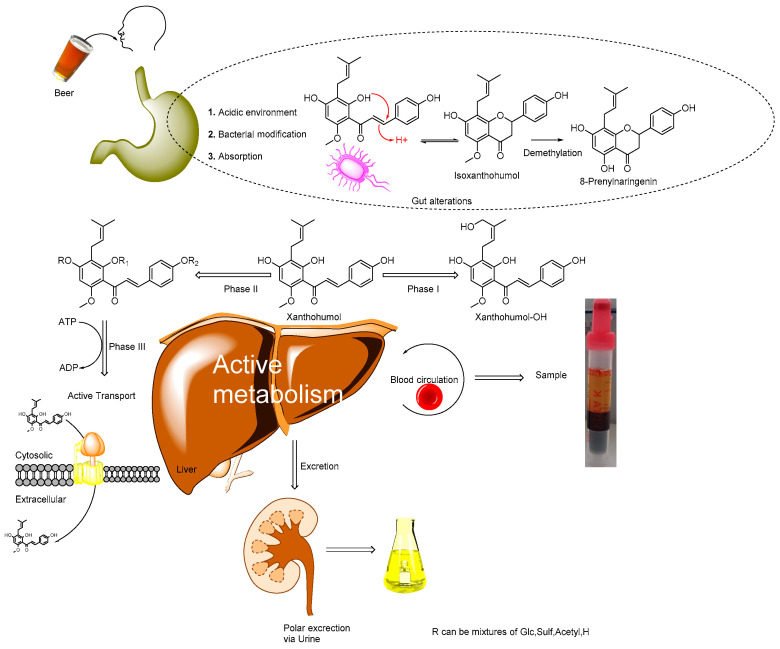
A generalised metabolic pathway of prenylated flavonoids from beer. Initial changes occur in the harsh environment of the gut, with additional metabolism by gut bacteria. Absorbed compounds that are changed in the gut (including xanthohumol) then make it to the liver, where active metabolism occurs.

**Figure 2 metabolites-12-00345-f002:**
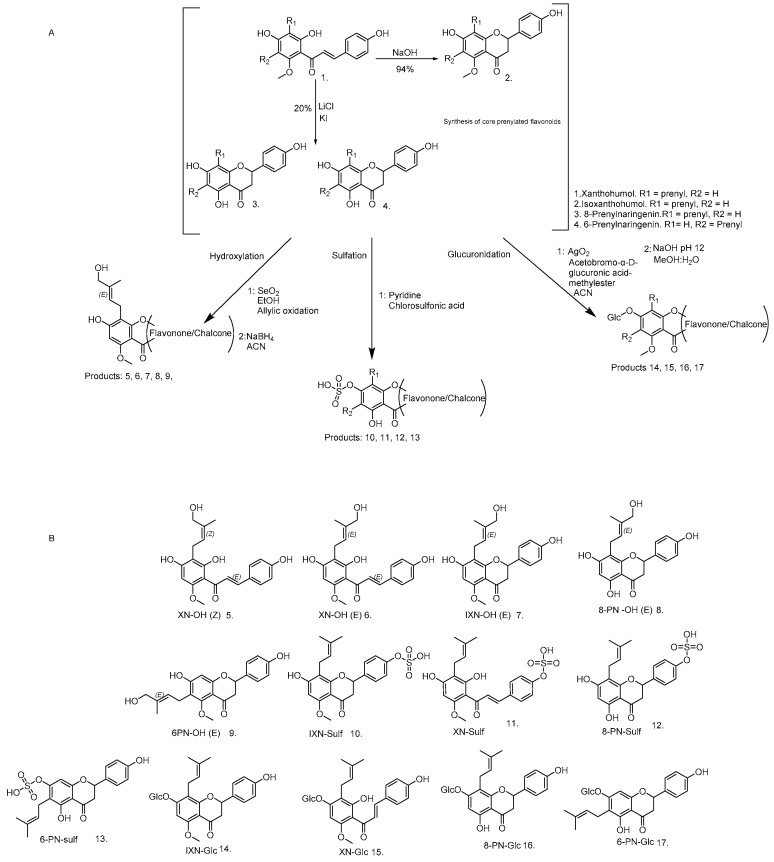
The generalised reaction scheme (**A**) for the synthesis of prenylated flavonoid metabolites. Products 1–4 served as the scaffold to further react into certain metabolites in either chalcone or flavanone form [18,20]. (**B**) The structures of each prenylated flavonoid synthesised (see Appendix A for detailed structure elucidation data).

**Figure 3 metabolites-12-00345-f003:**
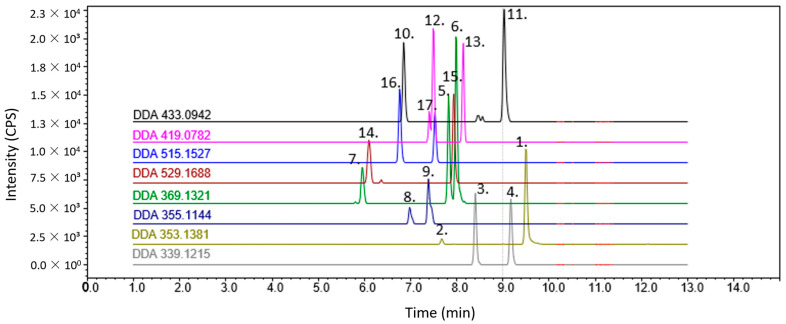
The DDA chromatogram (base shifted) of the analytical standards produced. Substances 1–17.

**Figure 4 metabolites-12-00345-f004:**
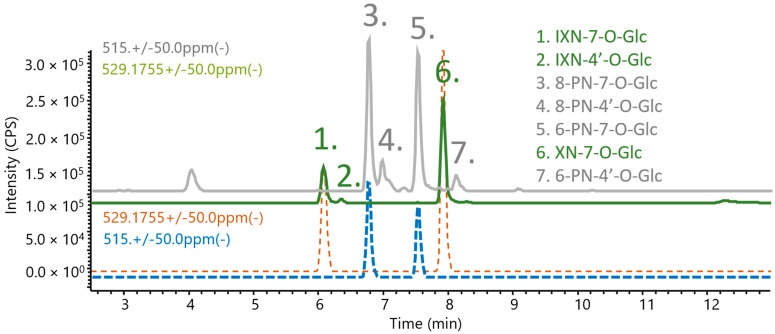
The identification of glucuronides after 60 min of XN consumption: co-chromatograms of the synthesised prenylated flavonoids (dashed blue, brown) and the compounds found in blood (green and gray).

**Figure 5 metabolites-12-00345-f005:**
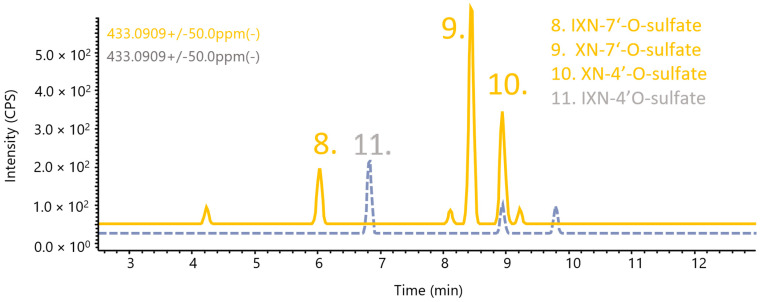
Co-chromatogram of the synthesised XN/IXN sulfates (gray) and analysis of human plasma (orange); compounds **8** and **9** were tentatively assigned.

**Figure 6 metabolites-12-00345-f006:**
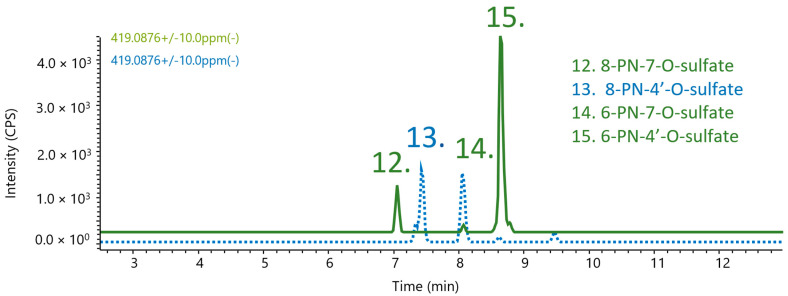
Co-chromatogram of the synthesised 8/6PN-sulfate, and then a human blood sample (green); compounds **12** and **15** were tentatively assigned.

**Table 1 metabolites-12-00345-t001:** The compounds found in human blood at different levels of identification according to Schymanski, E.L. (2014) [34].

Level 1: Confirmed Structure (Reference Standard)	Level 2: Probable Structure (Ms^2^)	Level 3: Tentative Structure (MS^2^)	Level 4: Molecular Formula	Level 5: Exact Mass of Interest (MS)
XN-7-O-GLc	-	IXN-4′O-Glc	-	-
IXN-7-O-Glc	-	8-PN-4′-O-Glc	-	-
6PN-7-O-Glc	-	6-PN-4′-O-Glc	-	-
8-PN-7-O-Glc	-	8-PN-7-O-sulfate	-	-
XN-4′O-Sulfate	-	6-PN-7-O-sulfate	-	-
6-PN-7-O-sulfate	-	6-PN-4′-O-sulfate	-	-

## Data Availability

The data presented in this study are available on request from the corresponding authors, as it has not been uploaded to an online database.

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
