# Peer review of "Synthesis of Human Phase I and Phase II Metabolites of Hop (Humulus lupulus) Prenylated Flavonoids"

_metabolites, 2022, doi:10.3390/metabo12040345_

Round 1

Reviewer 1 Report

The reviewed paper entitledhuman XN  "Synthesis of human phase I and phase II metabolites of hop (Humulus lupulus) prenylated flavonoids" describes the synthesis of reference standards for the identification of human xanthohumol (XN) metabolites. The synthesis products were analysed and compared to XN metabolites present in human blood. Several XN metabolites in human blood sample were identified on the basis of the synthesized reference standards.

The methods are clearly and in detail written. The relatively high number of syntheses and analysis methods of the synthesized products makes it difficult to present the results in a compact form and the authors have chosen more descriptive way by combining Results and Discussion sections. The study provides new information on how XN is metabolized in the human body and provides a basis for XN metabolism research. The article is in general very well written in excellent English. Some minor corrections are listed below (Minor comments).

Minor comments:

p. 3, r. 101 chrolosulfonic => Chlorosulfonic

p. 3 NaCO2 and AgO2, p. 5 H2O: number as subscript

p. 4, r. 127: Kamtha => Kamath

p. 4, r. 133: MgCl => MgCl2

p. 4, r. 157: 4ynthesized => please used the word "synthesised" uniformly throughout the manuscript

p. 6, r. 222: MEOH => MeOH

p. 6, r. 239: an excess "in"

p. 8, r. 266: hydroxl => hydroxyl

p. 8, 3.3. Please write UDPGA and UGT open when used for the first time

p. 13, acknowledgements: an excess "for the"

Author Response

Dear Reviewer,

We all would like to thank you for your very thorough review of the article "Synthesis of human phase I and phase II metabolites of hop (Humulus lupulus) prenylated flavonoids".

Especially noticing “Kamtha” which was very easy for us to skip over when reading the manuscript so often. I apologise that the track changes were not possible to show clearly the changes as the editor has explained that the layout of the article needs drastic changes even though we followed the author's guidelines (i.e. the complete article is changed). Therefore, we have modified the “entire” article and the track changes is applied and everything red. For the simplicity we added the recommended minor comments you have given. The text is nearly the same and we have applied all of the minor comments above – just the layout is different.

Another part is in the results in discussion which we added a small discussion at line 257-269:

“The description of XN metabolism is much more complicated than XN reaching cells and inducing certain bioactivities. Once consumed, XN is modified by a series of processors, including passive and non-passive routes resulting in the formation of many metabolites  (Figure 1). Furthermore, gut microbiota plays an important role [35]. Another factor that makes investigating the bioactivities of XN challenging to answer, is most in vivo bioactivity use mice models. The in vivo mice model for XN is prone to error as most mice do not have the essential human microbiota for the bioactivity effects of XN supplementation [36-38]. Although, some of the metabolites are probably bioactive as in vivo effects are reported from XN consumption [39,40]. Some metabolites might be rapidly excreted and suggest that absorption into cells is limited, although to answer this unequivocally, preliminary investigation using the synthesised metabolites with a Caco -2 absorption assay is necessary [41][9]”

You will also notice a small change in Figure 1 as we have added the structure of IXN and formation to 8-PN along with a diagram of phase III excretion.

We hope these changes are fine.

Best regards,

Lance

Reviewer 2 Report

The manuscript “Synthesis of human phase I and phase II metabolites of hop (Humulus lupulus) prenylated flavonoids " is interesting and well written. The manuscript describes a synthetic approach to synthesize reference standards for the identification of human xanthohumol metabolites in the human blood. This study can serve as a quality control examination of identified metabolites of xanthohumol that are in the blood circulatory system. I only have some minor remarks:

The authors can add a separate discussion part to discuss their results in the light of the previous reports, particularly XN is heavily metabolized in the body and thus, the previously reported bioactivity of XN was probably due to these metabolites. Further details on this crucial issue should be discussed comprehensively in a discussion part.

As revealed by the authors, different types of metabolites can be produced from XN. These metabolites differ in their drug-likeness properties, and hence, their accessibility to the cells and in turn, their bioactivity. For example, I believe that the sulfated derivatives are highly water-soluble and will be rapidly excreted from the body. Simple LogP calculation or ADME properties calculation can putatively discriminate between these metabolites in terms of their pharmacokinetics and pharmacokinetics. Such simple analysis could be added to the discussion part of a separate section.

Minor issues:

1- In Figure 1, please draw the structure of the cyclization process of XN in the stomach. Also, the structure resulted from phase III metabolism in the liver.

2- Line 104: correct NaCO2 to be NaCO2. Please check the whole manuscript for similar errors.

Finally, I should thank the authors for their efforts and for this valued and interesting piece of research.

Best Regards

Author Response

Dear Reviewer,

First, thank you very much for the kind comments, constructive criticism and great suggestions on the manuscript. We agree that the logP value calculation is a valid and important point, but making these points in this manuscript and calculations would be beyond the scope of this work and this may be something to consider for future experiments for example making use of these metabolites in e.g. Caco-2 cell culture and in vitro digestion (bioaccessibility experiments). However, I added the logP values of each metabolite into the supplementary information. We have additionally added the below text into the results and discussion to accommodate your statement and added further information on this matter. Lines 257-268

“The description of XN metabolism is much more complicated than XN reaching cells and inducing certain bioactivities. Once consumed, XN is modified by a series of processors, including passive and non-passive routes resulting in the formation of many metabolites  (Figure 1). Furthermore, gut microbiota plays an important role [35]. Another factor that makes investigating the bioactivities of XN challenging to answer, is most in vivo bioactivity use mice models. The in vivo mice model for XN is prone to error as most mice do not have the essential human microbiota for the bioactivity effects of XN supplementation [36-38]. Although, some of the metabolites are probably bioactive as in vivo effects are reported from XN consumption [39,40]. Some metabolites might be rapidly excreted and suggest that absorption into cells is limited, although to answer this unequivocally, preliminary investigation using the synthesised metabolites with a Caco -2 absorption assay is necessary [41][9]”

The comments from Figure 1:  The cyclisation is included and we have stated that phase III is the active transport across the cell membrane (efflux) and is included in the figure now. Additionally, I added that IXN can be demethylated in the figure.  

2 – We have fixed all the errors regarding the incorrect subscripts and changed many other words that were not consistent for example estrogen is oestrogen and metabolized -> metabolised.

A final comment: the editor has informed us that we have had to change the layout of the manuscript and I apologise that the track changes have completely taken over the entire manuscript we have done our best to keep the manuscript to the original.

Again thank you for the valid and informative review of the manuscript and I hope it is now more satisfactory,

Lance
